# Molecular Phylogeny of the Subfamily Notodontinae (Lepidoptera: Noctuoidea: Notodontidae)

**DOI:** 10.3390/insects16050526

**Published:** 2025-05-15

**Authors:** Muyu Guo, Qingliu Geng, Dandan Zhang

**Affiliations:** School of Ecology, Sun Yat-sen University, Shenzhen 518107, China; guomy25@mail2.sysu.edu.cn (M.G.); gengqliu@mail2.sysu.edu.cn (Q.G.)

**Keywords:** molecular phylogeny, mitochondrial protein-coding genes, low-coverage whole-genome sequencing

## Abstract

This study investigates the evolutionary history and phylogenetic relationships of the Notodontinae subfamily of Notodontidae. The aim is to clarify the classifications within this group and when the tribes diverged from one another. DNA sequences, mainly in the form of mitochondrial genes, from 57 species across 37 genera are analyzed, along with 78 related species within and outside the group for comparison. The morphological characteristics of male genitalia are also examined to support these findings. The results show that Notodontinae formed a distinct evolutionary group that appeared around 22.71 Ma in the Miocene, which then divided into three clades (or tribes) that split from one another between 22 and 19 Ma. By using fossil evidence to estimate timing, the study suggested that these moths first appeared around 23 million years ago. These findings help reconstruct the classification of Notodontidae moths, facilitating future research on their relationships and evolutionary history. This work provides a foundation for understanding biodiversity, conserving species, and tracing how traits and habitats have changed over millions of years.

## 1. Introduction

The family Notodontidae Stephens, 1829 (Lepidoptera: Noctuoidea), typically encompasses 8–18 subfamilies with fluctuating boundaries across taxonomic systems [1,2,3], comprising over 4400 species. The larvae are characterized by dorsal protrusions (“Notos” plus “odont”, meaning “back tooth”) and anal prolegs, earning them the name “prominent moths”. Key morphological adaptations include larval forked or spiny tails (as seen in Cerurinae), chemical defenses (such as formic acid secretion), and a boat-shaped posture in most adults. Host plant preferences span Salicaceae, Fagaceae, and Fabaceae, with larval gregariousness occasionally causing ecological impacts [4].

The subfamily Notodontinae (Notodontidae) was first proposed by Moore in 1882. Initially, eight genera were assigned to this subfamily: *Notodonta* Ochsenheimer, 1810, *Netria* Walker, 1855, *Antheua* Walker, 1855, *Ceira* Walker, 1865, *Pheosia* Hübner, 1819, *Sphetta* Walker, 1865, *Ichthyura* Hübner, 1819, and *Beara* Walker, 1866. The type genus *Notodonta*, with its designated type species *N. dromedarius*, has retained taxonomic validity since its original designation. However, subsequent taxonomic revisions have substantially redefined the composition of Notodontinae. For example, *Antheua* has been transferred to Phalerinae [1,5], *Sphetta* has been transferred to Platychasmatinae [3], *Ceira* has been recognized as the type genus of Ceirinae [6,7,8,9], and *Ichthyura* and *Beara* have been transferred to other families.

Neumoegen and Dyar (1894) classified *Notodonta*, *Nadata* Walker, 1855, *Symmerista* Hübner, 1821, *Cerura* Schrank 1802, *Clostera* Samouelle, 1819, *Gluphisia* Boisduval, 1828, *Pheosia*, *Peridea* Stephens, 1828, *Datana* Walker, 1855, and *Nystalea* Guenée, 1852, within Notodontinae at first [10], primarily based on adult synapomorphies such as forewing venation patterns (including the existence of accessory cell, whether the median vein of primaries is 3-branched, and whether veins 3 and 4 are stalked or separate) and larval caudal projections (forked or flagella-like, existence of warts and setae). However, in a 1897 revision, they redefined the classification by the absence of adult proboscis and larval secondary setae, transferring 57 genera (including *Notodonta*) out of Notodontidae to establish the family Ptilodontidae [11]. This revision was seen as controversial and later refuted.

Packard (1895) expanded Notodontinae to encompass genera such as *Notodonta*, *Pheosia*, *Odontosia* Hübner, 1819, *Nadata*, *Peridea*, *Hyperaeschra* Butler, 1880, *Ellida* Grote, 1876, and *Nerice* Walker, 1855, alongside taxa formerly placed in Nystalinae (such as *Dasylophia* Packard, 1864, and *Symmerista* Hübner, 1821), based on forewing (R_2_–R_5_ stalked, M_2_ closely approximated to M_3_, discal cell closed by M_2_–M_3_ alignment), hindwing (R_S_+R_1_ paralleling costa), as well as larval nutant tubercles [12]. In contrast, Matsumura (1925, 1929) focused on East Asian fauna, delimiting the subfamily to seven genera (*Notodonta*, *Spatalia* Hübner, 1819, *Phalera* Hübner, 1819, *Stauropus* Germar, 181*2*, *Uropyia* Staudinger, 1892 (=*Harpyia* Ochsenheimer, 1810), *Ptilophora* Stephens, 1828, and *Rosama* Walker, 1855) based on the bipectinate antennae and larval caudal bifurcations [6,7].

Draudt’s global revision of Notodontinae in 1932, encompassing 154 genera, formed a huge group. The diagnostic characteristics included the reduction or weakening of the forewing M_2_, the branching pattern of the R-series, the venation configuration within the hindwing discal cell, and the degree of sclerotization and presence of bifurcated structures in the uncus of the male genitalia [13]. Additionally, Forbes (1939, 1948) proposed a classification divided into five tribes (Gluphisiini, Notodontini, Nystaleini, Heterocampini, and Hemiceratini), tentatively using larval cephalic sclerotization and forewing R_5_ branching patterns [8,9]. Nevertheless, he expressed little satisfaction with this opinion.

Kiriakoff (1950) proposed a superfamily, Notodontoidea, comprising Dioptidae, Notodontidae, Thaumetopoeidae, and Thayetidae, which retained most genera within the Notodontinae and divided it into three tribes (Notodontini, Pygaerini, and Gluphisiini) [14]. His redefinition primarily relied on the morphology of male genitalia, particularly the shape and sclerotization of the uncus and gnathos, and larval morphology, specifically the presence of specialized spines on the eighth abdominal tergum. Decades later, Miller’s cladistic analysis (1991) reshaped the subfamily’s boundaries. By integrating taxa from Cerurinae, Gluphisinae, and Stauropinae, he identified two provisional clades, Notodontini and Dicranurini, though their monophyly had previously remained uncertain due to overlapping larval and adult traits with Heterocampinae (such as accessory cell, larval head capsule, and setae, which appeared as transitional characteristics) [1]. However, Miller’s reliance on larval ecology to distinguish Heterocampinae revealed the limitations of morphological apomorphies in resolving deep nodes, which were also present in Nakamura’s system (2007). He divided the subfamily into four tribes (Dudusini, Stauropini, Dicranurini, and Notodontini) based on pupal characteristics [15], with Dudusini obviously not belonging to this group.

Kobayashi and Nonaka (2016) proposed a reclassification of Notodontinae using single 28S rRNA gene sequences, analyzing them separately under maximum likelihood and Bayesian frameworks [3]. The study suggested a four-tribe system (Neodrymoniaini, Stauropini, Dicranurini, and Notodontini) comprising 70% of the Notodontidae samples, with the internodes receiving low statistical support (Bayesian posterior probabilities <0.8; bootstrap values <50), particularly for deeper divergences. The analysis by Kobayashi and Nonaka was the first molecular phylogenetic investigation of Notodontidae, with extensive taxon sampling that provided valuable insights into the classification of this family. However, insufficient morphological support left substantial room for further discussion.

Historically, the delimitation of Notodontinae from its closely related subfamilies, such as Dicranurinae, Stauropinae, Ptilodoninae, Heterocampinae, Ceirinae (partial), and Dudusinae (partial), has remained ambiguous due to extensive morphological homoplasy, leading to conflicting classifications in early taxonomic systems. Although Miller (1991) established a robust framework for Notodontidae classification by utilizing comprehensive morphological datasets and cladistic methodologies, it remains imperative to conduct deeper phylogenetic investigations. Subsequent molecular studies, such as that of Kobayashi and Nonaka (2016), attempted to reconstruct the family’s phylogeny using a single marker (28S rRNA), yet the low nodal support and absence of morphological evidence lowered the authority and persuasiveness of their argument. Consequently, critical issues persist in defining Notodontinae’s boundaries, necessitating phylogenomic approaches alongside extensive datasets to reconcile morphological synapomorphies with molecular synapomorphies.

To clarify the phylogenetic relationships within Notodontinae, this study attempted to assemble 13 protein-coding genes (PCGs) from the mitochondrial genomes of 135 samples (including 14 outgroups and 9 species from the public data platform NCBI) to form the backbone of the whole family, with 57 species belonging to 37 genera within Notodontinae. The phylogenetic tree was constructed mainly using maximum likelihood (ML). After confirming the monophyly of Notodontidae, the phylogenetic relationships within the subfamily Notodontinae were reconstructed and discussed. Based on phylogenetic results, the classification was supplemented and validated by a supporting dataset of orthologous genes (OGs) and characteristics of male genitalia, with the aim of clarifying the results using both molecular and morphological evidence. Moreover, divergence time estimation was conducted based on the phylogenetic tree.

## 2. Materials and Methods

### 2.1. Taxon Sampling

All specimens examined in this study were freshly collected from the field. Light trapping was employed to capture the specimens in their appropriate habitats. Upon capture, single or both legs (depending on the specimen size) were immediately excised and stored in 1.8 mL cryovials filled with pure ethanol for subsequent genomic work. The remaining specimen parts were mounted and spread as dried specimens for morphological analysis. In the laboratory, images of the dried specimens were captured using a Canon EOS 80D camera (Canon, Oita, Japan). The abdomens were then dissected to prepare genitalia slides, which were imaged using a Zeiss Axioscan 7 automated digital slide scanning system (Zeiss, Baden-Württemberg, Germany) for precise morphological identification.

### 2.2. Molecular Experiments

Following morphological identification, selected samples were retrieved from the cryovials for genomic DNA extraction using the CTAB method [16]. The extracted genomic DNA was then prepared for high-throughput sequencing. For library construction, 1 μg of genomic DNA was fragmented using a Covaris ultrasonicator to generate appropriately sized fragments. Magnetic bead-based selection was employed to enrich the fragments between 200 and 400 bp. The fragmented DNA was then end-repaired with A-tailing added at the 3′ end, followed by adapter ligation. The library was amplified via PCR, and the amplified products were purified using magnetic beads. The PCR products were then denatured into single-stranded DNA and circularized to form single-stranded circular DNA molecules. DNA nanoballs (DNBs) were generated, with any remaining linear DNA digested, yielding the final circularized library.

The qualified libraries were sequenced using combinatorial probe-anchor synthesis (cPAS) technology on the DNBseq platform, with a paired-end read length of 150 bp (PE150). For each sample, at least 4 Gb of clean data was obtained (with a minimum coverage of 10× genome size). Quality control of the clean data was conducted using SOAPnuke, and the quality scores of the reads were reported in Phred+33 format.

### 2.3. Phylogenetic Analyses

After obtaining the sequencing data, mitochondrial genome assembly was undertaken using the de novo strategy in MitoZ version 2.4 [17]. The assembled mitochondrial genome sequences were annotated using the online tool Mitos 2 in Galaxy version 2.1.9 [18], with the annotation results imported into Geneious Prime version 2024.07 for manual refinement to ensure the accuracy of the annotated regions. Following this, the 13 PCGs from each sample were extracted and imported into PhyloSuite version 1.2.3 [19] to begin the downstream analysis. The analytical workflow in PhyloSuite proceeded as follows: Sequences were aligned using MAFFT v7.313 [20], and poorly aligned regions were trimmed by utilizing Gblocks 0.91b [21]. The resulting alignments were concatenated into a supermatrix, followed by partitioning the scheme selection using PartitionFinder2 v2.1.1 [22] to identify the best-fit partitioning strategy and substitution model for the concatenated dataset. Phylogenetic trees were then constructed via ML, performed on IQ-TREE v1.6.8 [23], with ultrafast bootstrap approximation performed with 5000 replicates. Lastly, the resulting phylogenetic trees were visualized and refined using FigTree v1.4.4 to generate final graphical representations of the phylogenetic relationships.

Another dataset for the phylogenetic analysis of Notodontidae comprised OGs. Reference sequences for the OGs were downloaded from the OrthoDB v11.0 online database. The selection criteria included all 16 species within the Lepidoptera (*Leguminivora glycinivorella*, *Operophtera brumata*, *Hyposmocoma kahamanoa*, *Chilo suppressalis*, *Helicoverpa armigera*, *Plutella xylostella*, *Amyelois transitella*, *Spodoptera litura*, *Bombyx mori*, *Bombyx mandarina*, *Spodoptera frugiperda*, *Trichoplusia ni*, *Helicoverpa zea*, *Galleria mellonella*, *Galleria mellonella*, and *Ostrinia furnacalis*), excluding Papilionoidea, with filters set to “Present in all species” and “Single-copy in >80% species”. The alignment and trimming procedures were undertaken using PhyloAln v1.0.0 [24] and its associated scripts. Script alignseq.pl was deployed to perform codon-based multiple sequence alignment (MSA) for the reference sequences. Then, Trimal v1.5 [25] was used to trim the codons from each MSA. PhyloAln facilitated the alignment with reference sequences. To further refine the MSAs and remove sequences with an excessive number of unknown sites, trim_matrix.py was implemented, targeting columns with <90% “N” and rows with <10 (with “N” indicating unknown sites). The concatenation of all codon MSAs and filling in the gaps within the species were completed using connect.pl to generate partition files for subsequent phylogenetic analysis, with IQ-TREE construction undertaken with the same parameters as above.

### 2.4. Divergence Time Estimation

For divergence time estimation, BEAST v1.10.4 [26,27,28,29] was utilized, focusing on 13 PCGs of the mitochondrial genome sequences. The Notodontidae family was treated as monophyletic due to phylogenetic analyses, with five fossil calibration points applied (Table 1): *Plutellites tenebricus* from Plutellidae [30], *Eopyralis morsae* from Pyralidae [31], *Oligamatites martynovi* from Erebidae [32], *Tortricibaltia diakonoffi* from Tortricidae [33], and *Cerurites wagneri* from Notodontidae [34]. The codon positions were partitioned into two sets, positions 1+2 and position 3, with the TN93+I substitution model applied to *cox2*, while the GTR+Gamma+I model was selected for all other partitions based on the results from PartitionFinder2. An uncorrelated relaxed clock model was employed with lognormal distribution, while a Yule process was used as the tree prior. A randomly generated starting tree was selected for the tree model. The MCMC analysis was run for 60,000,000 generations, with log parameters for every 3000 generations. After the BEAST output was obtained, it was processed in TreeAnnotator v1.10.4, with the following settings: burn-in set to 20,000,000 stages, a posterior probability limit of 0.5, and the maximum clade credibility tree selected as the target tree. Node heights were set to median values.

## 3. Results

### 3.1. Dataset Construction and Phylogeny

#### 3.1.1. Monophyly of Notodontinae

Although both mitochondrial and OG datasets were employed, the conclusions were discussed based on the result from the mitochondrial dataset (the main tree) only, as this dataset possessed superior taxonomic comprehensiveness and species coverage. In contrast, the OG dataset exhibited less representativeness at the family level. In the main tree, samples consisted of 57 species from 37 genera within the ingroup of Notodontinae and 78 outgroups, among which 64 belonged to other subfamilies of Notodontidae. The 19 sequences listed in Table 2 were downloaded from Genbank. The matrix comprising 13 mitochondrial PCGs contained 10,980 bp (663 bp of *atp6*, 156 bp of *atp8*, 1,146 bp of *cob*, 1530 bp of *cox1*, 681 bp of *cox2*, 786 bp of *cox3*, 933 bp of *nad1*, 1,008 bp of *nad2*, 348 bp of *nad3*, 1266 bp of *nad4*, 285 bp of *nad4l*, 1653 bp of *nad5*, and 525 bp of *nad6*). Various data types were employed during the calculations, such as nucleotides (nt 12, 3, or nt 123) and translated amino acid sequences, as well as adjusted parameters for conventional bootstrapping or ultrafast-bootstrapping (only accepted for IQ-TREE) and specific outgroups. Although this study focused on reconstructing the phylogenetic relationships within Notodontinae, a basic familial backbone of Notodontidae was also provided to verify the monophyly of the subfamily and improve the accuracy of the divergence time calibration (Figure 1).

In the phylogenetic tree, the Notodontidae was assembled as a monophyletic group, exhibiting a structure of (Platychasmatinae, ((Pygaerinae, Dudusinae), (((Cerurinae, (Thaumetopoeinae, Phalerinae)), Ceirinae), Notodontinae))). The monophyly of Notodontinae was well supported, forming a sister group relationship with the ((Thaumetopoeinae, Phalerinae), (Cerurinae, Ceirinae)) clade.

According to the tree topology, three tribes were identified: Stauropini Matsumura, 1925 (Figure 1a), Notodontini Stephens, 1829 (Figure 1b), and Fentoniini Matsumura, 1929 (Figure 1c), with Notodontini as the sister group to (Stauropini, Fentoniini). The genus–tribe associations involved in this study are shown in Table 3.

The second dataset, assembled from OGs, incorporated 589 filtered loci (919,493 bp in total) for supplementary validation (supporting tree, Figure 2). The taxon sampling was slightly different from that of the mitochondria dataset, comprising 52 species of 34 genera from Notodontinae and 78 outgroups, among which 55 belong to other subfamilies of Notodontidae. Four outgroup data were downloaded from Genbank (Table 2). Within Notodontidae, the subfamilial topology was resolved as (Platychasmatinae, ((Pygaerinae, Dudusinae), (Ceirinae, ((Cerurinae, (Thaumetopoeinae, Phalerinae)), Notodontinae)))). The key distinction was that the main tree displayed the sister relationship of Notodontinae to (Cerurinae, (Thaumetopoeinae, Phalerinae)), whereas Ceirinae was positioned externally to this quartet of subfamilies. Nevertheless, the basal branching pattern (Platychasmatinae, (Pygaerinae, Dudusinae)) and the terminal placement of Notodontinae remained highly congruent across both datasets. Within Notodontinae, the tribal relationships mirrored the main tree, with discrepancies restricted to generic-level groupings within Notodontini (discussed below).

#### 3.1.2. **Stauropini** Matsumura, 1925

Figure 3, Figure A1 (1–6)

The composition of Stauropini was taxonomically stable in both phylogenetic results, with all five genera forming two internal clades. One clade consisted of the sister group *Cnethodonta* and the tribe’s type genus *Stauropus*, with strong support for the monophyly of *Stauropus*. In the other clade, *Somera* was the sister group to the remaining two genera, *Netria* and *Syntypistis*.

Diagnosis.

The genitalia of Stauropini’s males with weakly developed uncus and gnathos, usually short and non-sclerotized; valva mildly sclerotized, often with delicate sclerotized sella; saccus papillary or elongated; phallus longer than the length from uncus to saccus, especially in *Stauropus*.

#### 3.1.3. **Notodontini** Stephens, 1829

Figure 4, Figure A1 (7–15)

Notodontini was the largest group within the subfamily, with 22 genera clustered. In accordance with the tree topology and internal nodes, it was divided into one basal branch plus three lineages. The basal branch was *Stauroplitis*, as the sister group to the other Notodontini group. The first lineage consisted of (*Zaranga*, (*Omichlis*, *Formofentonia*)), followed by a clade consisting of ((*Hagapteryx*, *Himeropteryx*), *Pheosia*), with *Pheosia* as the sister group to the other two genera. In the final lineage, *Ptilodon* was the sister group to all other taxa, followed by *Homocentridia*, then a hierarchical grouping composed of (*Epodonda*, *Semidonta*). A branch consisting of *Lophocosma* and *Allodontoides* diverged next, followed by (*Nerice*, *Togepteryx*). Within the subsequent clade, *Euhampsonia* was the sister group to (*Rachiades*, *Notodonta*), with the latter including the type genus *Notodonta* (type species: *N. dromedarius*). The final divergence placed *Metriaeschra* as the sister group to (*Pheosiopsis*, (*Acmeshachia*, *Peridea*)), forming the terminal branch of the clade.

In the supporting tree (the OG dataset), Notodontini comprised three primary lineages. The first lineage clustered *Stauroplitis* together with four genera as (*Stauroplitis*, (*Zaranga*, (*Omichlis*, *Formofentonia*))), while the second lineage aligned (*Hagapteryx*, (*Himeropteryx*, *Pheosia*)), which was identical to the main tree topology. The genera *Ptilodon*, *Pheosiopsis*, *Acmeshachia,* and *Peridea* retained the same placement as in the main tree within the third lineage, but topological incongruence emerged among the remaining ten genera: *Nerice* and *Allodontoides* formed a clade, succeeded by a quartet branch (*Epodonta*, ((*Semidonta*, *Homocentridia*), *Euhampsonia*)), followed by nested clusters of (*Togepteryx*, *Lophocosma)* and (*Metriaeschra*, *Rachiades)*. Critically, both datasets exhibited subfamilial-level concordance, reinforcing the proposed classification system.

Diagnosis.

Notodontini male’s genitalia uncus weakened, approximately half the length of the tegumen, proportionably smaller than that in other tribes; gnathos short, often double-hook-shaped; valva stripe-shaped, without editum, distal gradually narrow; costa slightly sclerotized, bearing 1–2 prominent distal half, occasionally with similar processes on sacculus; sacculus weakly sclerotized; phallus strongly developed, distal sclerotized, fork-shaped or with process.

#### 3.1.4. **Fentoniini** Matsumura, 1929

Figure 5, Figure A1 (16–21)

Fentoniini included nine genera in this study, with very high support for each internal node. This tribe was topologically divided into two clades, with one comprising two sister groups, (*Neopheosia*, *Fentonia*) and (*Parachadisra*, *Betashachia*). In the other clade, *Neodrymonia* was the first to diverge, followed by a branch consisting of (*Pseudofentonia*, (*Mesophalera*, (*Disparia*, *Libido*))).

Diagnosis.

Uncus of Fentoniini’s male genitalia strong, usually prominent and/or forkshaped, more developed than the other two tribes’; gnathos strongly developed and sclerotized; valva membranous, often oval, marginal smooth, base of costa with slightly sclerotized process; juxta broad, often platy or peltate; saccus U-shaped, middle bottom sometimes concave; phallus strong and thick, sometimes with developed spine at phallus tip, densely with cornuti.

### 3.2. Divergence Time Estimation

Based on the results of the BEAST analysis of divergence time estimation (Figure 1), the crown group origin of Notodontidae was estimated to be approximately 30.68 Ma in the Oligocene (95% highest posterior density, HPD = 33.42–27.75 Ma). The subfamily Notodontinae diverged from its sister group at 22.71 Ma (HPD= 24.81–20.54 Ma) in the Miocene. Within Notodontinae, the earliest split occurred between the lineages leading to Stauropini and Fentoniini at 22.32 Ma (HPD= 24.44–20.23 Ma). Subsequently, the tribe Stauropini originated at around 21.66 Ma (HPD = 23.83–19.53 Ma), with Notodontini formed contemporaneously at around 21.45 (HPD = 23.60–19.10 Ma). Lastly, the youngest tribe, Fentoniini, originated from a split within the Stauropini lineage at 19.20 Ma (HPD= 21.70–16.63 Ma).

## 4. Discussion

The taxonomy of Notodontinae has long been challenging due to its diversity, as well as its complicated and ambiguous morphological characteristics. Due to the diversity and variability within this subfamily, relying solely on morphology makes it difficult to propose a clear, stable, and convincing classification for subordinate taxa. In this study, molecular evidence supports the classification of three distinct tribes: Stauropini Matsumura, 1925, Notodontini Stephens, 1829, and Fentoniini Matsumura, 1929. All three tribes consistently cluster as monophyletic groups, forming a stable tree structure.

Several tribes that were historically classified under Notodontinae have been phylogenetically reassessed. Kiriakoff’s Pygaerini [14] and Nakamura’s Dudusini [15] were recovered as monophyletic subfamilies, in alignment with a recent consensus [1,3]. In contrast, Heterocampini in Forbes’ study [8,9] (or Heterocampinae in most early studies) and Dicranurini in the studies of Miller and Kobayashi and Nonaka [1,3,15] (referred to as Dicranurinae in all of Schintlmeister’s studies) have exhibited paraphyly, leading to their taxonomic invalidation. The other two tribes not addressed here were transferred outside to be independent clades in early studies, including Nystaleini in Forbes [8,9], which Miller elevated to subfamily rank, and Gluphisiini, which was invalidated concurrently (originally proposed as Gluphisiinae by Packard [12], and later demoted to tribe status by Forbes and Kiriakoff [8,9,14]). The current study lays a foundation for a new phylogenetic perspective on the family level beyond Notodontinae, although the positions of Hemiceratini, Nystaleinae, and Dioptinae within Notodontidae remain unresolved due to insufficient sampling, likely because of their restricted Neotropical and Nearctic distribution [1,3].

According to Bayesian inference and fossil calibration, Notodontidae originated from around 30.68 Ma, and Notodontinae subsequently diverged at 22.71 Ma. The most recent common ancestor (MRCA) of Stauropini and Fentoniini originated at 22.32 Ma, predating the subsequent divergence of their respective tribal lineages. Stauropini was the earliest diverging tribe, emerging at 21.66 Ma, followed rapidly by the split of Notodontini at 21.45 Ma. Subsequently, Fentoniini split from the ancestral lineage of Stauropini at 19.20 Ma, suggesting a secondary diversification event within this lineage.

### 4.1. **Stauropini** Matsumura, 1925

The subfamily Stauropinae was established by Matsumura in 1925, with *Stauropus* as the type genus. Miller (1991) subsequently downgraded it to tribal rank (Stauropini) under Heterocampinae, comprising four genera: *Stauropus*, *Cnethodonta*, *Schizura,* and *Harpyia*, while placing *Quadricalcarifera* (a synonym of *Syntypistis*) in Dicranurini [1]. Nakamura (2007) later transferred Stauropini to within Notodontinae based on pupal morphology [15], a classification also supported by Kobayashi and Nonaka [3]. Schintlmeister proposed replacing Stauropinae with Dicranurinae (*Dicranura* as the type genus), including all genera adopted in the current Stauropini, but his regional monographs only provided simple morphological justifications [2,5,35,36].

Historically, Dicranurinae (or Dicranurini) encompassed genera now classified in other subfamilies [1,2,3,5,15,35,36,37], such as *Furcula*, *Harpyia* (in Cerurinae), *Stauropus* (in Notodontinae), *Ptilophora* (in Notodontini), and *Gluphisia* (in Pygaerinae) [1,3,37]. However, molecular evidence has revealed the paraphyletic nature of traditional Dicranurinae, with *Furcula*, *Cerura*, *Harpyia*, *Kamalia*, and *Neocerura* forming a monophyletic Cerurinae, while *Stauropus* (the type genus of Stauropini), *Cnethodonta*, *Netria*, and *Syntypistis* remain individually within Notodontinae. Due to the absence of *Dicranura* samples in the current study and the need for taxonomic stability, Stauropini was retained as a valid tribe within Notodontinae, consistent with its type genus *Stauropus*.

### 4.2. **Notodontini** Stephens, 1829

Forbes (1939, 1948) first formally identified tribal divisions within Notodontinae, defining Notodontini to include six genera: *Notodonta*, *Ellida*, *Nadata*, *Peridea*, *Pheosia*, and *Odontosia* [8,9]. Miller (1991) later reassigned *Peridea*, *Nadata*, and *Ellida* to Phaleriniae, whereas *Notodonta*, *Pheosia*, and *Odontosia* remained within Notodontini [1]. Kobayashi and Nonaka’s molecular study (2016) supported Miller’s placement (of *Notodonta*, *Nadata*, *Peridea*, *Pheosia*, and *Odontosia* within Notodontini), but transferred *Ellida* to Periergosinae [3]. Traditionally, Ptilodotinae (=Ptilophorinae Matsumura, 1929) was related to Notodontinae in terms of classification, but it performed as a paraphyletic group in molecular studies, with *Ptilodon*, *Hagapteryx*, *Togopteryx*, *Himeropteryx*, *Semidonta*, *Allodontoides*, and *Epodonta* transferred from Ptilodotinae to Notodontinae [3,5]. In contrast to the latest phylogenetic study, *Stauroplitis*, *Zaranga*, *Omichlis*, and *Formofentonia*, which previously belonged to Stauropini, were reassigned to Notodontini.

In the current study, two genera, *Euhampsonia* and *Zaranga*, which have usually been assigned to Dudusinae, were found to share more characteristics with Notodontini. In terms of male genitalia, the uncus of Dudusinae is often bifurcated or apically incised, the valva rhomboid and membranous, and the saccus usually elongate-triangular. However, *Euhampsonia* and *Zaranga* exhibit a single-protrusion uncus, narrow-elongate and strongly sclerotized valva, and flattened saccus, suggesting a separation from Dudusinae. Furthermore, the medium-sized uncus, processes of costa, plate-shaped juxta, and flattened saccus confirmed their taxonomic placement within Notodontini (Figure 4B,I). The genera belonging to previous Ptilodontinae also shared common characteristics with other species within Notodontini, such as the weakened uncus, hook-shaped gnathos, prominent distal half on costa, stripe-shaped valva, and flattened saccus (Figure 4D–G,J).

The phylogenetic topology of Notodontini showed marked differences in comparison to the dataset of mitochondrial PCGs and OGs, with one reflected in the basal four branches, which were clustered into a monophyletic clade in the supporting tree, separated from the remaining Notodontini by extended divergence times. However, this grouping lacked corroborations in both the alternative dataset and the synapomorphies in the external morphology or genitalia. Nevertheless, the extended branch lengths and divergence times observed in *Stauroplitis*, *Zaranga*, and the (*Omichlis*, *Formofentonia*) branches suggested potential novel taxonomic arrangements within the basal Notodontini, warranting expanded taxon sampling in future studies to clarify their evolutionary status. Another difference is related to the distal radial branches. The phylogeny revealed a rapid, near-simultaneous diversification event, yielding a species-rich group. Morphological assessments (Figure 4F–M) identified overlapping characteristic states between these lineages, including uncus size and sclerotized intensity, shape or contour of valva, and costa sclerotized with distal projections, implying that the evolutionary rates of current molecular markers may be inadequate for resolving such rapid radiations. More crucial and appropriate molecular markers should be identified in future studies so that the actual phylogenetic relationships can be clarified.

### 4.3. **Fentoniini** Matsumura, 1929

Matsumura (1929) first erected Fentoninae in his classification of 11 subfamilies but did not list any genera or diagnoses [7]. The current study indicates that this monophyletic group consists of nine genera, including *Fentonia* and *Neodrymonia*, type genus of Fentoninae and Neodrymoniaini **syn. nov.**, respectively. Since Neodrymoniaini was erected by Kobayashi and Nonaka in 2016 [3], much later than Fentoninae, the latter was downgraded to tribal rank within Notodontinae, with the scientific name Fentoniini accepted for this clade.

This tribe exhibited a strong monophyly in the current analysis, so the validity was confirmed by the current study. Four genera were newly assigned: *Neopheosia*, *Fentonia*, *Betashachia*, and *Parachadisra*. Kiriakoff (1968) combined *Pheosia atrifusa* with *Parachadisra* [38]; it was later synonymized with *Chadisra* by Schintlmeister and assigned to Notodontinae [5], whereas Kobayashi (2016) placed it within Periergosinae. In contrast, the results of this study revealed *Pheosia*, *Chadisra*, and *Parachadisra* as independent lineages, thereby supporting the retention of *Parachadisra* as a valid genus. The remaining three genera were transferred from Heterocampinae [3].

Although the phylogenetic and divergence time estimation results suggested that Fentoniini and Stauropini should seemingly belong to one tribal clade, the morphological evidence clarified the independence of each tribe. Species within Stauropini exhibit incomplete forewing coverage of the hindwings during rest, exposing the patterned scales along the costal margin of hindwings as thick, “roof-like” configurations, with the mimetic strategies resembling bird droppings or decaying matter. In contrast, the Fentoniini species’ hindwings only possess common hair and are completely covered by the forewings. The patterns and comparatively thinner body align more with bark- or leaf-mimicry adaptations. In the male genitalia, Stauropini exhibit a more specialized and complex configuration of the eighth sternite, characterized by the absence of socii, valva with apical or sacculus-basal projections, a ventrally projecting and elongated saccus, and a significantly elongated phallus lacking cornuti. Fentoniini display a relatively simplified morphology of the eighth sternite, well-developed socii, a broad valva with pronounced projections on the basal costa, a U-shaped saccus, and a robust, moderately long phallus bearing cornuti. Given that the morphological disparities significantly exceeded the variation typically observed within tribal-level classifications, these two clades were recognized as distinct tribes.

### 4.4. Conclusions

The current analysis incorporated a substantial number of notodontid samples outside of Notodontinae, allowing for a credible conclusion regarding tribal-level relationships and the assignment of genera. Compared to the latest system that was erected by Kobayashi and Nonaka [3], some Stauropini lineages were transferred into Notodontini, Neodrymoniaini was synonymized with Fentoniini, and the validity of Dicranurini was derecognized. Within Notodontini, *Zaranga*, *Omichlis*, and *Formofentonia* were transferred as new members from Stauropini. Additional reassignments pertained to Fentoniini, among which *Parachadisra* was reinstated as a valid genus synonymous with *Chadisra,* transferred from Periergosinae, and *Neopheosia* and *Fentonia* were reassigned from the previous Heterocampinae.

Given that this study focuses on a single subfamily in clarifying its internal phylogenetic relationships, it does not deeply address the validity or taxonomic justifications of other subfamilies. Some historically problematic paraphyletic subfamilies, such as Dicranurinae and Ptilodoninae, are tentatively treated as invalid units in this study. Whether these subfamilies are truly legitimate remains a question for future studies, using expanded sample sizes.

## Figures and Tables

**Figure 1 insects-16-00526-f001:**
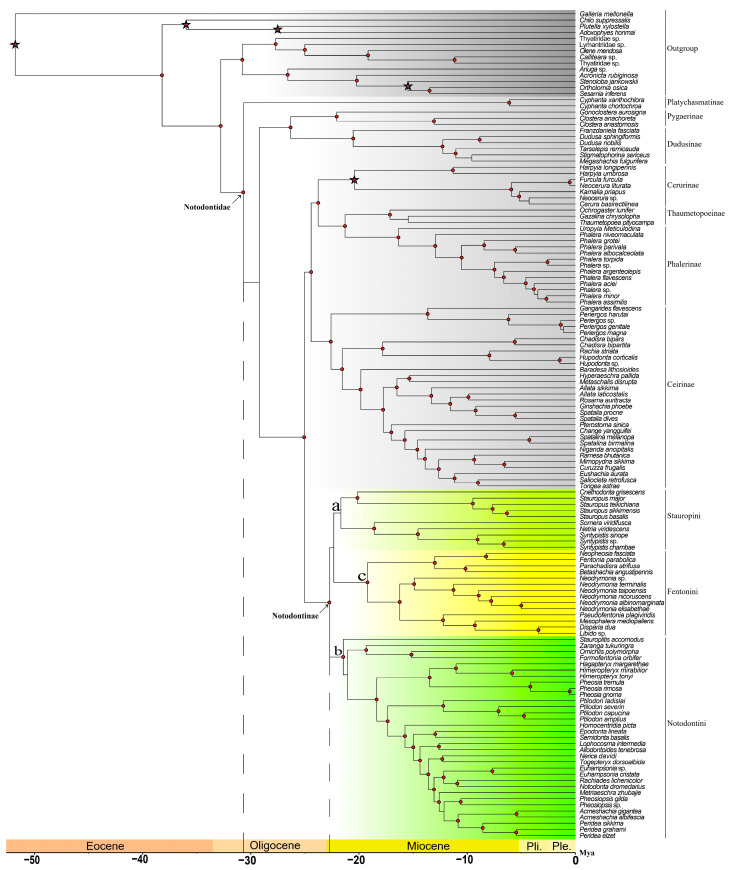
Phylogenetic results by mitochondrial genome dataset (main tree). (**a**): Stauropini, highlighted by aqua; (**b**): Notodontini, highlighted by green; (**c**): Fentoniini, highlighted by yellow. Circle-shaped nodes’ supporting value > 90; time scale showed as horizontal axis; asterisks indicate fossil calibration points.

**Figure 2 insects-16-00526-f002:**
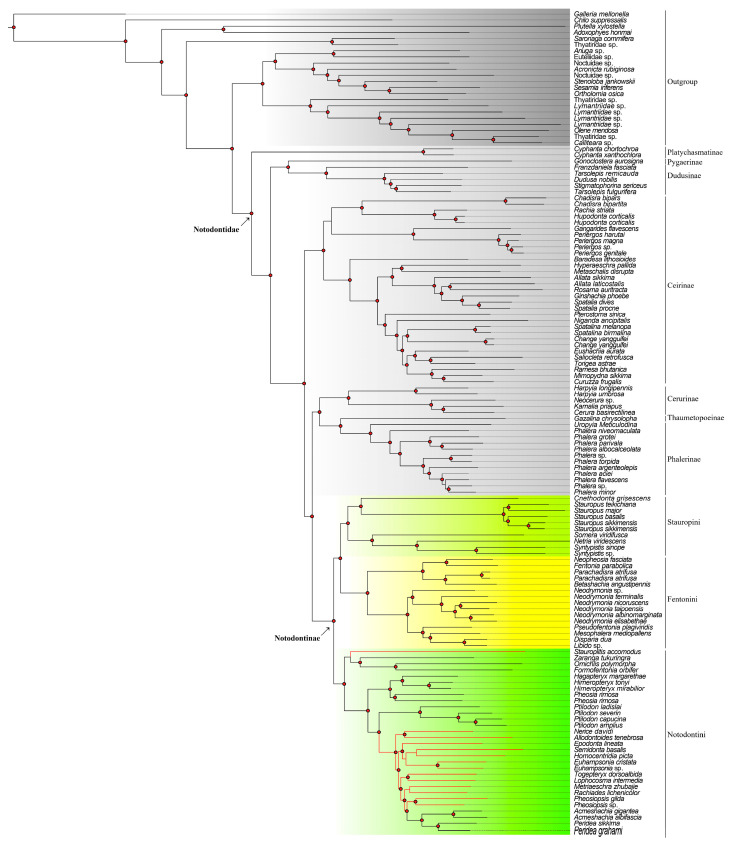
Phylogenetic results from the OG dataset (supporting tree). Stauropini highlighted by aqua, Notodontini highlighted by green, Fentoniini highlighted by yellow. Circle-shaped nodes’ supporting value > 90; red branches show a difference compared to the main tree.

**Figure 3 insects-16-00526-f003:**
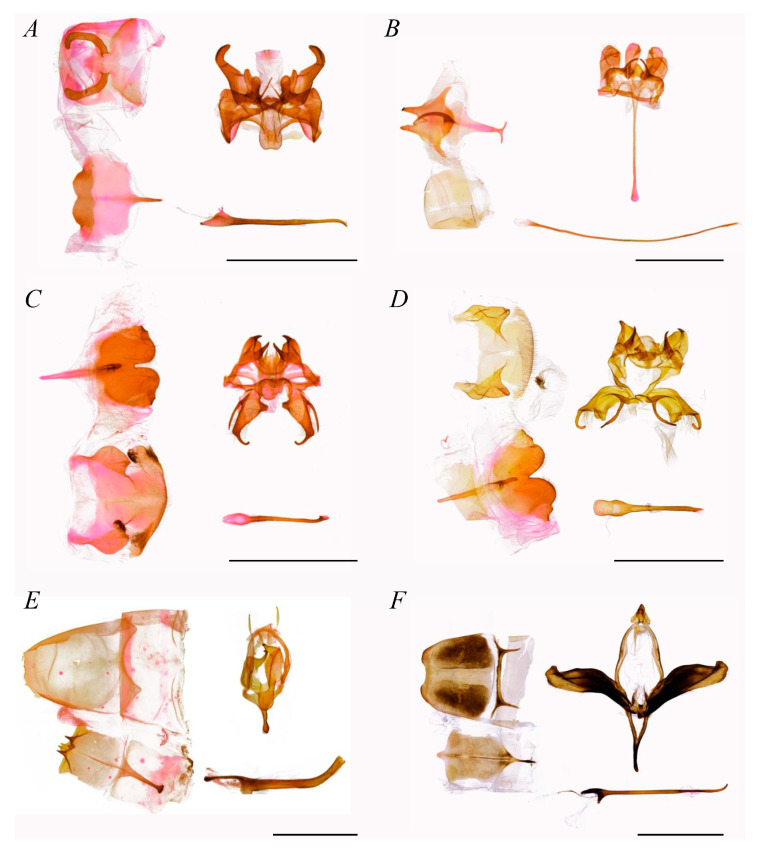
Male genitalia of Stauropini (scale bar = 5 mm). (**A**) Stauropus major, (**B**) Stauropus basalis, (**C**) Stauropus sikkimensis, (**D**) Stauropus sikkimensis, (**E**) Netria viridescens, (**F**) Syntypistis sinope.

**Figure 4 insects-16-00526-f004:**
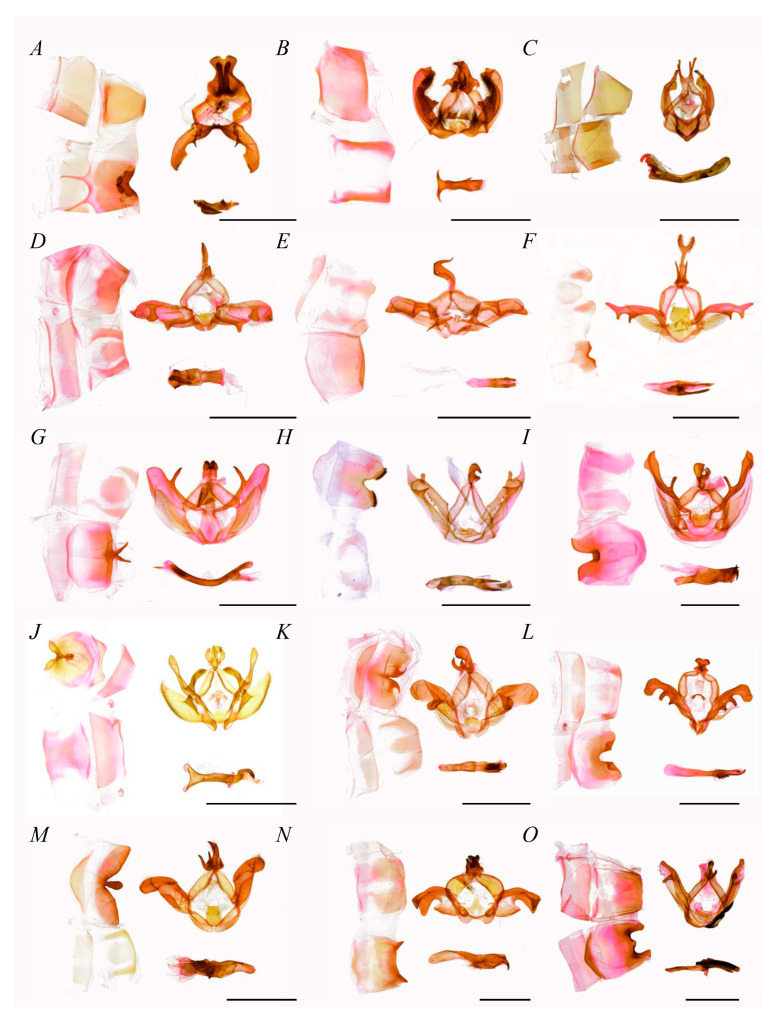
Male genitalia of Notodontini (scale bar = 5 mm). (**A**) *Stauroplitis accomodus*, (**B**) *Zaranga tukuringra*, (**C**) *Formofentonia orbifer*, (**D**) *Himeropteryx mirabilior*, (**E**) *Ptilodon ladislai*, (**F**) *Allodontoides tenebrosa*, (**G**) *Epodonta lineata*, (**H**) *Homocentridia picta*, (**I**) *Euhampsonia cristata*, (**J**) *Togepteryx dorsoalbida*, (**K**) *Lophocosma intermedia*, (**L**) *Metriaeschra zhubajie*, (**M**) *Pheosiopsis gilda*, (**N**) *Acmeshachia gigantea*, (**O**) *Peridea sikkima*.

**Figure 5 insects-16-00526-f005:**
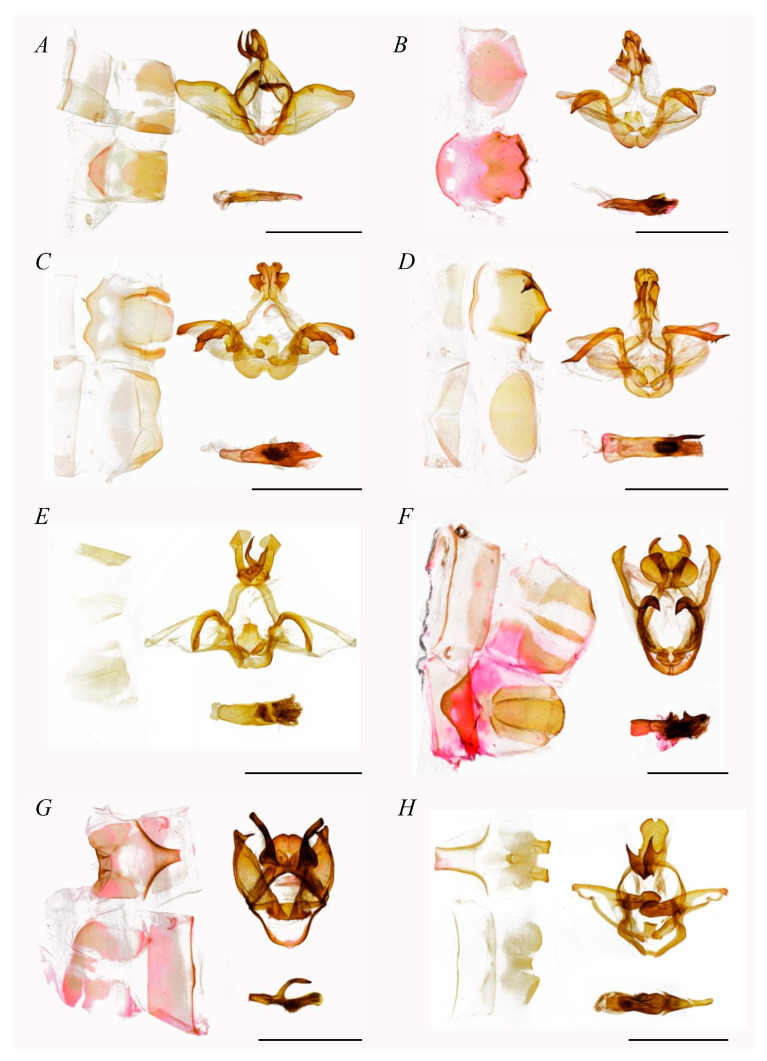
Male genitalia of Fentoniini (scale bar = 5 mm). (**A**) *Parachadisra atrifusa*, (**B**) *Neodrymonia taipoenis*, (**C**) *Neodrymonia nicoruscens*, (**D**) *Neodrymonia albinomarginata*, (**E**) *Pseudofentonia plagiviridis*, (**F**) *Mesophalera amica*, (**G**) *Disparia dua*, (**H**) *Libido* sp.

**Table 1 insects-16-00526-t001:** Fossil calibrations used in Notodontidae divergence time estimation.

Taxon	Family	PBDB (Ma)	Fossil Type	Location
*Plutellites tenebricus*	Plutellidae	38.0–33.9	Baltic Amber	Russia
*Eopyralis morsae*	Pyralidae	56.0–47.8	Rock	Denmark
*Oligamatites martynovi*	Erebidae	28.1–23.0	Rock	Kazahkstan
*Tortricibaltia diakonoffi*	Tortricidae	38.0–33.9	Baltic Amber	Russia
*Cerurites wagneri*	Notodontidae	3.6–2.588	Rock	Germany

**Table 2 insects-16-00526-t002:** Sequences downloaded from Genbank.

Species Name	Accession No.	Species Name	Accession No.
*Adoxophyes honmai*	NC_008141	*Cnethodonta grisescens*	NC_068541
	GCA_005406045.1	*Dudusa sphingiformis*	MW788876
*Chilo suppressalis*	NC_015612	*Furcula furcula*	OU452271
	GCA_902850365.2	*Neocerura liturata*	NC_062182
*Galleria mellonella*	KT750964	*Ochrogaster lunifer*	NC_011128
	GCA_026898425.1	*Peridea elzet*	NC_062111
*Plutella xylostella*	NC_025322	*Phalera assimilis*	OP784764
	GCA_932276165.1	*Pheosia gnoma*	FR989925
*Sesamia inferens*	NC_015835	*Pheosia rimosa*	NC_061645
	GCA_037179545.1	*Pheosia tremula*	HG995428
*Clostera anachoreta*	NC_034740	*Syntypistis chambae*	NC_062123
*Clostera anastomosis*	NC_041140	*Thaumetopoea pityocampa*	NC_053256

**Table 3 insects-16-00526-t003:** Tribe–genus associations within Notodontinae (bolded genera indicate newly added members in this study).

Stauropini	Notodontini	Fentoniini
*Stauropus*, *Somera*, *Cnethodonta*, *Netria*, *Syntypistis*	***Stauroplitis*, *Zaranga*, *Omichlis*, *Formofentonia***, *Hagapteryx*, *Himeropteryx*, *Pheosia*, *Ptilodon*, *Homocentridia*, *Allodontoides*, *Lophocosma*, *Semidontoides*, *Epodonta*, *Togepteryx*, *Nerice*, *Euhampsonia*, *Rachiades*, *Notodonta*, *Metriaeschra*, *Pheosiopsis*, *Achmeshachia*, *Peridea*	***Neopheosia*, *Fentonia*, *Parachadisra***, ***Betashachia***, *Pseudofentonia*, *Mesophalera*, *Disparia*, *Libido*, *Neodrymonia*

## Data Availability

The data are openly available in GenBank at https://www.ncbi.nlm.nih.gov/genbank/.

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
