# Peer review of "Molecular Phylogeny of the Subfamily Notodontinae (Lepidoptera: Noctuoidea: Notodontidae)"

_insects, 2025, doi:10.3390/insects16050526_

Round 1
Reviewer 1 Report
Comments and Suggestions for Authors
The submitted MS represents a comprehensive study of a diverse and taxonomically challenging group of Notodontid moths. The authors have provided new insights into the systematics of the Notodontinae subfamily using modern techniques (e.g. phylogenomic approach based on both, mitochondrial and nuclear DNA), which significantly improve the data presentation and quality of the obtained results. The original data presented here undoubtedly worthy of publication in MDPI “Insects”. However, the MS requires further English improvement and editing prior to acceptance for publication. I strongly recommend the MS to be checked and revised by a native English speaker. Some (not all) corrections are provided in the attached file.
Please, also add scale bars to genitalia dissections images figured in the text.

The MS requires further English improvement and editing prior to acceptance for publication. I strongly recommend the MS to be checked and revised by a native English speaker.
Author Response
Comments 1: The submitted MS represents a comprehensive study of a diverse and taxonomically challenging group of Notodontid moths. The authors have provided new insights into the systematics of the Notodontinae subfamily using modern techniques (e.g. phylogenomic approach based on both, mitochondrial and nuclear DNA), which significantly improve the data presentation and quality of the obtained results. The original data presented here undoubtedly worthy of publication in MDPI “Insects”. However, the MS requires further English improvement and editing prior to acceptance for publication. I strongly recommend the MS to be checked and revised by a native English speaker. Some (not all) corrections are provided in the attached file.
Please, also add scale bars to genitalia dissections images figured in the text.
Response 1: Thank you for pointing this out. We agree with this comment. The manuscript has undergone thorough checking and revision by a native English-speaking editor. The genitalia plates have been updated to color version with expanded specimens and scale bars (page 9, 10, 12).
Comments 2: The MS requires further English improvement and editing prior to acceptance for publication. I strongly recommend the MS to be checked and revised by a native English speaker.
Response 2: The manuscript has been revised in accordance with the annotations provided in your PDF file.
The reason for prioritizing mitochondrial genomic data in our discussion lies in its higher taxonomic coverage and sequence completeness compared to the nuclear gene dataset, which exhibited a few matrix incompleteness. The topological incongruences between the two datasets was principally observed within the Notodontini clade, some expanded discussion has been incorporated in Section 4.2 (page 14).
Current study followed the latest family-level phylogenetic classification proposed by Kobayashi & Nonaka (2016). We included representative species from majority Notodontinae-associated units historically recognized (e.g., Pygaerini, Dudusini, Heterocampini, Dicranurini). Regarding unsampled groups, two achievements have been widely recognized since Miller (1991): (1) Nystaleini has been elevated to subfamilial rank independent without Notodontinae; (2) Gluphisiini has been synonymized and abolished as a valid tribe. Our phylogenetic reconstruction therefore maintains representativeness within the currently accepted taxonomic framework (page 13, line 364-377).
Your expert guidance has significantly enhanced our study more logical clearer, we remain deeply grateful for your scholarly insights and patience throughout the review process!
Reviewer 2 Report
Comments and Suggestions for Authors
The authors provide a molecular phylogeny of Notodontinae, a subfamily for which classification has traditionally been contentious. Their analysis is based on 13 protein-coding mitochondrial genes, obtained through high throughput sequencing, as well as another dataset comprised of orthologous genes obtained from a public database. This dataset is highly suited for this purpose. Taxon sampling for the study is impressive. In addition to including 57 species traditionally encompassed within Notodontinae, they also included 78 outgroups within and outside of Notodontidae to provide a preliminary phylogeny for the family and assess which taxa should rightfully be included in Notodontinae. Three tribes within Notodontinae are recovered by their data sets. Characteristics of the male genitalia are briefly described and photographed for each tribe. Since morphological characterization of the genitalia is not detailed, it is of limited use. Divergence times for each tribe were calculated based on calibration with the fossil record. The study is thorough and provides a robust phylogeny for Notodontinae and a framework for further research within Notodontidae classification. These aspects merit publication as the data presented are both reliable and useful.
There are minor concerns with the figures:
Figure 1: It would be more helpful if the colors used to indicate Stauropini and Notodontini were more different, since they appear to be almost identical. It is hard to see the circle shaped notes on the tree. Would it be possible to make the circles bigger or otherwise more prominent?
Figures 3, 4 and 5: The images appear to be out of focus, and it is difficult to see detail in relevant characters. Some structures appear overstained or oversaturated. It might be more useful to provide color images.
Comments on the Quality of English LanguageThe manuscript could benefit from rather extensive language editing, as sentences are not always clear. I have tried to correct some language concerns in this review, but others remain.
Line 7: “and subfamily tree” can be deleted as it is redundant
Line 9: The term “DNA sequences” might be more appropriate in this context
Line 12: It is unclear who “their” is referring to here
Line 16: Sentence should read “These findings helped reconstruct the classification…”
Line 23: The meaning of the phrase “together with 78 outgroups within/outside Notodontidae” is unclear. An alternative phrasing could be: “…which included 57 species belonging to 37 genera within Notodontinae, together with 64 other species within Notodontidae and 14 outgroups.”
Line 26: Could read “…was performed as a supporting analysis”
Line 37: The keywords Notodontidae and Notodontinae are already included in the title and can be omitted here.
Line 63: Notodonta should be italicized
Line 80: Is it unclear what is meant by “satisfaction”
Line 95: Should read “…while the Dudusini obviously did not belong to this group”
Lines 103-104: Should read “However, insufficient morphological support left substantial room for further discussion.”
Line 120: Should read: “…whole family, with 57 species belonging to 37 genera within Notodontinae.”
Line 125: Should read: “…characteristics of the male genitalia…”
Line 177: It is unclear what is meant here. What does the 16 refer to? There are far more than 16 lepidopteran families.
Line 192: “due to phylogenetic analyses”. I am not sure what this phrase refers to.
Line 224: “monophly should read “monophyletic”
Line 225: Platychasmatinae is misspelled.
Line 243: Notodontidae should not be italicized
Line 259: It might be useful to state how taxonomic stability is defined.
Line 269: Genus names should be italicized.
Line 320: “estimation” is misspelled
Line 344: Not a complete sentence.
Line 348: Should this be “demoted” instead of “degraded”? Also occurs in line 360.
Line 384: Should read “Notodontinae”
Line 396: “notodontid” should not be capitalized.
Author Response
Comments 1: Figure 1: It would be more helpful if the colors used to indicate Stauropini and Notodontini were more different, since they appear to be almost identical. It is hard to see the circle shaped notes on the tree. Would it be possible to make the circles bigger or otherwise more prominent?
Figures 3, 4 and 5: The images appear to be out of focus, and it is difficult to see detail in relevant characters. Some structures appear overstained or oversaturated. It might be more useful to provide color images.
Response 1: Thank you for pointing this out. We agree with this comment. The circle shaped nodes have been enlarged, and the genitalia plates have been updated to color version with expanded specimens and scale bars (page 7, 8, 9, 10, 12).
Comments 2: The manuscript could benefit from rather extensive language editing, as sentences are not always clear. I have tried to correct some language concerns in this review, but others remain.
Line 7: “and subfamily tree” can be deleted as it is redundant
Line 9: The term “DNA sequences” might be more appropriate in this context
Line 12: It is unclear who “their” is referring to here
Line 16: Sentence should read “These findings helped reconstruct the classification…”
Line 23: The meaning of the phrase “together with 78 outgroups within/outside Notodontidae” is unclear. An alternative phrasing could be: “…which included 57 species belonging to 37 genera within Notodontinae, together with 64 other species within Notodontidae and 14 outgroups.”
Line 26: Could read “…was performed as a supporting analysis”
Line 37: The keywords Notodontidae and Notodontinae are already included in the title and can be omitted here.
Line 63: Notodonta should be italicized
Line 80: Is it unclear what is meant by “satisfaction”
Line 95: Should read “…while the Dudusini obviously did not belong to this group”
Lines 103-104: Should read “However, insufficient morphological support left substantial room for further discussion.”
Line 120: Should read: “…whole family, with 57 species belonging to 37 genera within Notodontinae.”
Line 125: Should read: “…characteristics of the male genitalia…”
Line 177: It is unclear what is meant here. What does the 16 refer to? There are far more than 16 lepidopteran families.
Line 192: “due to phylogenetic analyses”. I am not sure what this phrase refers to.
Line 224: “monophly should read “monophyletic”
Line 225: Platychasmatinae is misspelled.
Line 243: Notodontidae should not be italicized
Line 259: It might be useful to state how taxonomic stability is defined.
Line 269: Genus names should be italicized.
Line 320: “estimation” is misspelled
Line 344: Not a complete sentence.
Line 348: Should this be “demoted” instead of “degraded”? Also occurs in line 360.
Line 384: Should read “Notodontinae”
Line 396: “notodontid” should not be capitalized.
Response 2: We sincerely appreciate your meticulous corrections, and the manuscript has been comprehensively revised accordingly.
Your expert guidance has significantly enhanced our study more logical clearer, we remain deeply grateful for your scholarly insights and patience throughout the review process!
Reviewer 3 Report
Comments and Suggestions for Authors
Overall, there are a few related issues that the authors should address. First, authors should make sure that they have applied the most critical tests to ensure the conclusion of monophyly for each tribe sampled in the study. Please give some other external diagnosis for each tribe to fit the proposed hypothesis by this study or previous study. I certainly that is very important for tribe-level classification in the subfamily Notodontinae as well. The most important thing I concerned is that the number of sequenced taxa in this study were not clearly indicated either in Materials and Methods section and Results section. Does the dataset include any sample sequenced by other previous studies? In addition, I assume that they used same sequencing raw data to assemble sequences of both mitogenome and orthologous genes, but why authors only used tree inferred by mitogenomic data to interpret the phylogeny of Notodontinae instead of both dataset of mitogenome and orthologous genes?
Some comments and suggestions are attached in PDF file.

I strongly suggest to have a native English speaking read the manuscript.
Author Response
Comments 1: First, authors should make sure that they have applied the most critical tests to ensure the conclusion of monophyly for each tribe sampled in the study. Please give some other external diagnosis for each tribe to fit the proposed hypothesis by this study or previous study. I certainly that is very important for tribe-level classification in the subfamily Notodontinae as well. The most important thing I concerned is that the number of sequenced taxa in this study were not clearly indicated either in Materials and Methods section and Results section. Does the dataset include any sample sequenced by other previous studies? In addition, I assume that they used same sequencing raw data to assemble sequences of both mitogenome and orthologous genes, but why authors only used tree inferred by mitogenomic data to interpret the phylogeny of Notodontinae instead of both dataset of mitogenome and orthologous genes?
Response 1: Thank you for pointing this out. We agree with this comment. Some external diagnosis and the distinctions among each tribes have been supplemented in section 3.1, 4.2 and 4.3. Data downloaded from Genbank were noted in table 2, together with accession numbers (page 5, 14, 15).
The mitochondrial genomic dataset and orthologous genes matrix were analyzed separately rather than concatenated due to significant discrepancies in taxonomic coverage and sequence completeness. Specifically, the mitochondrial dataset demonstrated higher species representation and genomic integrity, whereas the OGs matrix exhibited insufficient taxonomic sampling and gene coverage for reliable concatenated phylogenetic reconstruction. Our future studies will integrate newly collected specimens with expanded taxon sampling to enable more robust phylogenetic reconstructions.
Comments 2: Some comments and suggestions are attached in PDF file.
Response 2: We sincerely appreciate your meticulous corrections marked in the PDF file. The manuscript has been comprehensively revised accordingly. The genitalia plates have been updated to color version with expanded specimens and scale bars (page 9, 10, 12).
Comments 3: I strongly suggest to have a native English speaking read the manuscript.
Response 3: The manuscript has undergone thorough checking and revision by a native English-speaking editor.
Your expert guidance has significantly enhanced our study more logical clearer, we remain deeply grateful for your scholarly insights and patience throughout the review process!
Reviewer 4 Report
Comments and Suggestions for Authors
The manuscript is very interesting, but it took few weeks to prepare the review.
The phylogenetic system of the subfamily Notodontinae, which the authors construct in their work, differs greatly from the generally accepted system of Eurasian Notodontinae [Schintlmeister, 2008], based on the system of Miller, 1991, but with modifications [Schintlmeister, 2008: 8]. The Schintlmeister’s [2008] system should also be analyzed in the introduction, but this was not done.
There must be some justification for what morphological features, in addition to DNA, the subfamily Notodontinae is distinguished on. It is on this basis that we can understand why the genera Zaranga and Euhampsonia were removed from Dudusinae and transferred to Notodontinae (Notodontini).
First, the subfamily Notodontidae must be outlined to understand why genera usually assigned to other subfamilies were included there (Zaranga, Euhampsonia, for example). At least it is unclear why they were even included in the tribe Notodontini?
The significant similarity of the proposed phylogenetic trees based on mitochondrial and nuclear genes speaks for their objectivity. However, it would be nice to see additional arguments (not based on DNA!) in the article showing the necessity of transferring Zaranga and Euhampsonia from Dudusinae to Notodontinae-Notodontini, as well as the inclusion of Ptilodontinae in Notodontinae-Notodontini.
According to the presented dendrograms, the genus Hupodonta was transferred to Ceirinae from Notodontinae (like Rosama, Spatalia were transferred from Pygaerinae). I would like to know what reasons there were for such a transfer, besides DNA.
Based on the figured trees, it is unclear why the authors propose to separate the tribe Neodrymoniaini from Stauropini in Notodontinae, but do not separate the most basal clades Stauroplitis, Zaranga+Omichis+Formofentonia into separate tribes? Especially since they branch off significantly earlier than Neodrymoniaini and Stauropini separate. At least, if the authors include Stauroplitis in Notodontini, then Stauropini and Neodrymoniaini should be combined.
Additionally, Neodrymoiaini Kobayashi, 2012 is much younger than Fentoniini Matsumura, 1929: 78 (at least, the genus Fentonia is included in the tribe Neodrymoniaini proposed by the authors). So, the oldest name for Neodrymoniaini should be Fentoniini.
After the above comments have been corrected, the manuscript can be published.
Author Response
Comments 1: There must be some justification for what morphological features, in addition to DNA, the subfamily Notodontinae is distinguished on. It is on this basis that we can understand why the genera Zaranga and Euhampsonia were removed from Dudusinae and transferred to Notodontinae (Notodontini).
First, the subfamily Notodontidae must be outlined to understand why genera usually assigned to other subfamilies were included there (Zaranga, Euhampsonia, for example). At least it is unclear why they were even included in the tribe Notodontini?
The significant similarity of the proposed phylogenetic trees based on mitochondrial and nuclear genes speaks for their objectivity. However, it would be nice to see additional arguments (not based on DNA!) in the article showing the necessity of transferring Zaranga and Euhampsonia from Dudusinae to Notodontinae-Notodontini, as well as the inclusion of Ptilodontinae in Notodontinae-Notodontini.
Response 1: Thank you for pointing this out. We agree with this comment. Detailed morphological evidence have been supplemented in Section 4.2. We have provided additional comparisons of genitalia structures highlighting the differences between Zaranga, Euhampsonia, and Dudusinae, as well as their shared characteristics with Notodontini. Additionally, morphological evidence supporting the incorporation of Ptilodontinae into Notodontini has been included (page 14).
Comments 2: According to the presented dendrograms, the genus Hupodonta was transferred to Ceirinae from Notodontinae (like Rosama, Spatalia were transferred from Pygaerinae). I would like to know what reasons there were for such a transfer, besides DNA.
Response 2: The classification we followed was the latest family-level phylogenetic work of Kobayashi & Nonaka (2016), in which the positions of Hupodonta, Rosama and Spatalia had already been confirmed by molecular evidences. Since our study focused on tribe-level within Notodontinae, the remain subfamilies were not discussed in the manuscript. It’ an interesting and necessary for our future investigation, sincerely thank for your suggestion!
Comments 3: Based on the figured trees, it is unclear why the authors propose to separate the tribe Neodrymoniaini from Stauropini in Notodontinae, but do not separate the most basal clades Stauroplitis, Zaranga+Omichis+Formofentonia into separate tribes? Especially since they branch off significantly earlier than Neodrymoniaini and Stauropini separate. At least, if the authors include Stauroplitis in Notodontini, then Stauropini and Neodrymoniaini should be combined.
Response 3: We have observed remarkable tribal differences between Neodrymoniaini syn. nov. and Stauropini on characters of both adult and genitalia, the detailed discussion has been added in section 4.3. However, since there were insufficient evidence supporting the independence for basal four genera (Stauroplitis, Zaranga, Omichis, Formofentonia) from Notodontini, we conservatively maintain the remains within Notodontini, and separate the other two tribes simoutainously (page 15, lines 461-476).
Comments 4: Additionally, Neodrymoiaini Kobayashi, 2012 is much younger than Fentoniini Matsumura, 1929: 78 (at least, the genus Fentonia is included in the tribe Neodrymoniaini proposed by the authors). So, the oldest name for Neodrymoniaini should be Fentoniini.
Response 4: We gratefully acknowledge this insightful nomenclatural critique, and have systematically replaced Neodrymoniaini Kobayashi, 2016 with the senior synonym Fentoniini Matsumura, 1929 throughout the revised manuscript (page 11, 15).
Your expert guidance has significantly enhanced our study more logical clearer, we remain deeply grateful for your scholarly insights and patience throughout the review process!